# Functional Comparison of Three Chitinases from Symbiotic Bacteria of Entomopathogenic Nematodes

**DOI:** 10.3390/toxins16010026

**Published:** 2024-01-03

**Authors:** Da-Jeong Son, Geun-Gon Kim, Ho-Yul Choo, Nam-Jun Chung, Young-Moo Choo

**Affiliations:** 1Department of Applied Bioscience, Dong-A University, Busan 49315, Republic of Korea; dajeongson@jbio.or.kr; 2Division of Research and Development, Jinju Bioindustry Foundation, Jinju 52839, Republic of Korea; 3Division of Research and Development, Nambo Co., Ltd., Jinju 52840, Republic of Korea; kgoni02@naver.com (G.-G.K.); chooka7@daum.net (H.-Y.C.)

**Keywords:** chitinase, Entomopathogenic nematode, *Xenorhabdus nematophilia*, *Xenorhabdus hominickii*, *Photorhabdus temperata*

## Abstract

*Xenorhabdus* and *Photorhabdus,* bacterial symbionts of entomopathogenic nematodes *Steinernema* and *Heterorhabditis*, respectively, have several biological activities including insecticidal and antimicrobial activities. Thus, XnChi, XhChi, and PtChi, chitinases of *X. nematophila*, *X. hominickii*, and *P. temperata* isolated from Korean indigenous EPNs *S. carpocapsae* GJ1-2, *S. monticolum* GJ11-1, and *H. megidis* GJ1-2 were cloned and expressed in *Escherichia coli* BL21 to compare their biological activities. Chitinase proteins of these bacterial symbionts purified using the Ni-NTA system showed different chitobiosidase and endochitinase activities, but N-acetylglucosamidinase activities were not shown in the measuring of chitinolytic activity through N-acetyl-D-glucosarmine oligomers. In addition, the proteins showed different insecticidal and antifungal activities. XnChi showed the highest insecticidal activity against *Galleria mellonella*, followed by PtChi and XhChi. In antifungal activity, XhChi showed the highest half-maximal inhibitory concentration (IC_50_) against *Fusarium oxysporum* with 0.031 mg/mL, followed by PtChi with 0.046 mg/mL, and XnChi with 0.072 mg/mL. XhChi also showed the highest IC_50_ against *F. graminearum* with 0.040 mg/mL, but XnChi was more toxic than PtChi with 0.055 mg/mL and 0.133 mg/mL, respectively. This study provides an innovative approach to the biological control of insect pests and fungal diseases of plants with the biological activity of symbiotic bacterial chitinases of entomopathogenic nematodes.

## 1. Introduction

*Xenorhabdus* and *Photorhabdus* are symbiotically and pathologically associated with the entomopathogenic nematodes (EPNs), *Steinernema* and *Heterorhabditis*, respectively [1]. The symbiotic relationship between EPN and bacterium begins with infection of an insect host with free-living, third-stage infective juvenile EPNs which house the mutualistic symbiotic bacterium in its intestine.

Once the infective juvenile enters the insect host through natural openings (mouth, anus, or spiracles) and penetrates into the hemocoel, the infective juvenile releases the bacterial cells from its intestine in the insect’s hemocoel, and symbiotic bacterium produces a range of secondary metabolites killing the host within 48 h by septicemia [2,3,4]. Recently, insecticidal proteins, as well as secondary metabolites of *Xenorhabdus* and *Photorhabdus*, have been targeted for their potential use in agricultural-pest-management investigation of the virulence mechanism [5,6,7]. The representative insecticidal proteins of *Xenorhabdus* include Xpt [8], Txp40 toxin [9], XaxAB [10], XnGroE [11], and PirAB [12]. *Photorhabdus* include a wide range of insecticidal proteins including multiunit toxin complexes (Tc), Photorhabdus insect-related (PirAB) toxins, XaxAB, and Photox binary toxins, Makes caterpillar floppy (Mcf), Photorhabdus virulence cassettes (PVC), Photorhabdus insecticidal toxin (Pit), and a ubiquitous Txp40 toxin [13,14,15,16,17]. In addition to the insecticidal proteins without catalytic activity of symbiotic bacteria, digestive enzymes that degrade an insect body are produced by bacteria to provide food for both bacteria and nematodes [18]. However, bacterial enzymes of EPNs associated with insecticidal activities have rarely been investigated.

Chitin, the second most abundant biopolymer in nature after cellulose, is a polymer composed of repeating units of β-1,4-N-acetylglucosamine (GlcNAc). Chitin serves as the main structural component of the extracellular matrix and is found in organisms, including insect exoskeletons, fungal cell walls, crustacean shells, and nematode eggshells [19,20]. Thus, chitinase’s key enzyme has received increasing attention as biopesticide for the control of insect pests and fungal diseases because chitin synthesis is performed using a range of organisms, including fungi and insects, and key enzyme in biosynthesis [21,22]. Chitinase is a Glycoside hydrolase (GH) that acts to degrade chitin using hydrolyzing glycosidic bonds [23], and it is classified into two main groups, chitinase (EC3.2.1.14) and β-acetylhexosaminidase (EC 3.2.1.52), officially called endochitinase and exochitinase, respectively, depending on the products produced during the hydrolysis process [24,25]. Endochitinase cleaves into the chitin chain at an internal site. Exochitinase contains two subcatogories, called chitobiosidases (EC 3.2.1.29) and β-N-acetylglucosaminidases (EC 3.2.1.30) that are now included with (EC 3.2.1.52), β-N-acetylhexosaminidase [24]. While chitobiosidases catalyze progressive release of di-acetylchitobiose from a terminal non-reducing end, β-N-acetylglucosaminidases cleave oligomeric products, such as (GlcNAc)_2_, (GlcNAc)_3_, (GlcNAc)_4_, obtained transforming endochitinase into monomers of N-actyl glucosamine (GlcNAc) [24,26]. These enzymes are grouped into different Glycoside Hydrolase (GH) Families namely GH18, 19 and 20, based on the amino acid similarity of chitinase from various organisms [27]. In particular, as chitinases belong to the GH18 family, they are widely distributed in almost all organisms and were found to be involved in many physiological processes including tissue degradation, developmental regulation, pathogenicity, and immune defense [28]. These chitinases from a variety of microorganisms which facilitate the invasion of pathogens by causing structural changes in the peritrophic membrane of insects, which is primarily composed of chitin, promoting the accessibility of the substrate for the pathogen into the haemocoel, and leading to the interception of nutrient absorption in the midgut [23,29]. Furthermore, chitinases can facilitate the binding process of toxins to specific receptors in the midgut epithelium of insects and enhance the insecticidal activity of entomopathogenic bacteria [30]. *Bacillus thuringiensis* chitinases (BtChi) have been previously reported to have a contribution to pathogenicity through synergistic effects in combination with other components including Cry proteins [24]. In bacterium *Yersinia entomophaga* MH96 isolated from diseased grass grub larva of *Costelytra zealandica* (Coleoptera: Scarabaeidae), the 3D structure of Tc toxins includes two putative chitinases (Chi1 and Chi2) which are essential for complex formation [31]. In addition, two chitinase (chi60 and chi70) genes were found in the toxin complex locus of *X. nematophila* [31] and the corresponding proteins were shown to be vital for the insecticidal activity against *Helicoverpa armigera* in the toxicity of Tc toxins [23]. Moreover, chitinases are able to inhibit the elongation and growth of mycelia and spore germination of fungi [32], that is, chitinase Chi2A and chitin binding protein (CBP) in the secondary cells of *P. luminescens* are necessary to inhibit the growth of *Fusarium graminearum* [33]. Thus, chitinases could be potential and effective virulence factors for the biological control of insect pests and plant pathogenic fungi.

*Xenorhabdus* and *Photorhabdus* show high similarity between clades, as 16S rRNA (16S rDNA) sequences indicate a close phylogenetic relationship between these two genera [34,35,36], but differ in the life cycle and in the pathogenic mechanisms including evasion of the insect immune system and expression of virulence factors [36]. To date, only comparative analyses of insecticidal activity has been performed for *Xenorhabdus* and *Photorhabdus* between strains of the same species or different genera [37,38]. However, factors accounting for the different pathogenicity among them have not been clearly investigated. In the recent comparative analysis of genes related to pathogenicity at different genus levels, insecticidal activities were different even in the same pathogenic protein [39]. This suggests the possibility that even the same pathogenic protein may have different biological activities, e.g., due to amino acid variability. Accordingly, chitinases are important toxic and antifungal factors against insect larvae and plant pathogenic fungi [32,33,40]; nevertheless, symbiotic bacterial chitnases of EPNs have not been studied except *X*. *nematophila* and *P*. *luminescens.* Therefore, chitinases of *X*. *nematophila* (GeneBank access number: OR724704), *X*. *hominickii* (GeneBank access number: OR724706), and *P*. *temperata* (GeneBank access number: OR724705), isolated from Korean indigenous EPNs *S*. *carpocapsae* GJ1-2, *S*. *monticolum* GJ11-1, and *H*. *megidis* GJ1-2 were cloned and expressed to compare biological activity. In addition, Phylogenetic analysis and multiple sequence alignment were performed to characterize the chitinase genes. Chitinolytic activity of their chitinases was measured to compare insecticidal and antifungal activity against *Galleria mellonella*, *F*. *oxysporum,* and *F*. *graminearum*.

## 2. Results

### 2.1. Sequencing and Phylogenetic Analysis of Chitinases of Xenorhabdus and Photorhabdus Strains Isolated from Korean Indigenous EPNs

Sequence analysis of the chitinase genes confirmed that chitinases of *X. nematophila* (XnChi), *X. hominickii* (XhChi), and *P. temperata* (PhChi) were of the GH18 family, but their length of polypeptides were different, i.e., 648, 498, and 637 aa, respectively (Appendix A).

In the multiple sequence alignment of chitinases of *X. nematophila*, *X. hominickii*, and *P. temperata*, including previously reported bacteria *Pseudomonas chloriraphis* B25, *Cronobacter sakazakii* wls2261, and *Y. entomophaga* MH96, chitinases of these six bacteria contained a GH18 catalytic domain, chitinase insertion domain (CID), chitin binding site (CBDs), and active site. Although sequences of the GH18 catalytic domain and the CID were significantly different among them, the CBDs and active site of chitinase were highly conserved. However, XhChi showed a relatively large difference due to its short length compared to the other five bacteria sequences (Figure 1).

In addition, chitinases of XnChi, XhChi, and PtChi did not belong to the same clade as those of previously studied *Pseudomonas*, *Bacillus*, and *Serratia* in the phylogenetic analysis. The same genus of EPN symbiotic bacteria were generally grouped into relatively close clades, but XnChi, XhChi, and PtChi distinctly formed their respective clades (Figure 2). Such tree topology indicates lineage-specific gene duplication and losses.

### 2.2. Cloning, Expression and Purification of Recombinant Chitnases

XnChi, XhChi, and PtChi were cloned and expressed using *Escherichia coli* expression system to compare their biological functions. The prominent bands of the recombinant proteins for XnChi, XhChi, and PtChi on the SDS–PAGE were detected at ca. 76 kDa, 59 kDa, and 75 kDa, respectively. The highest protein expression level was observed at 24 h after IPTG induction (Figure 3a). Because high-levels of protein expression could lead to inclusion body formation [41], samples were used at 18 h after IPTG induction, which showed similar expression levels (Figure 3a). Recombinant proteins were purified using Ni-NTA superflow resin under native conditions (Appendix A). In order to fully characterize the XnChi, XhChi and phChi, the eluted fractions were concentrated and desalted using an Amicon Ultra centrifugal filter (30 kDa cut-off). The purified recombinant chitinases showed the single individual bands with 76 kDa, 59 kDa, and 75 kDa on SDS–PAGE (Figure 3b).

### 2.3. Chitinolytic Activity Assay with Recombinant Chitinases

Chitinase is classified into exochitinase and endochitinase according to enzymatic action on chitin substrate. XnChi, XhChi, and PtChi showed chitiobiosidase, which is an exochitinase, and endochitinase activities in a concentration-dependent manner on the substrate, respectively. However, β-1,4-N-acetylglucosaminidases activity was not observed. At a concentration of 1000 ng/µL, which exhibited the highest activity of chitinase, chitobiosidase activity of XhChi was 2.4-fold and 2.1-fold higher than that of XnChi and PtChi, respectively. However, endochitinase activities of XnChi and PtChi were higher than those of XhChi by 0.5-fold and 0.7-fold, respectively (Figure 4).

### 2.4. Evaluation of Insecticidal Activity of Chitinase against Galleria mellonella

XnChi, XhChi, and PtChi were highly toxic against *G. mellonella*. Although mortality was different depending on the chitinase concentration, larval mortality was significantly higher at the rate of 40 µg (Figure 5). The median lethal time (LT_50_) value was calculated to compare insect mortality. LT_50_ values of XnChi, XhChi, and PtChi were ca. 16, 18, and 17 at the 40 μg, respectively (Table 1). In addition, chitinase affected the development of *G. mellonella*, and was not limited to a specific stage (Appendix A). *Galleria* larvae developed into adults in the control, whereas chitinase-fed larvae did not developed to pupae or adults. Most of them were dead in the larval or pupal stage (Appendix A).

### 2.5. Assessment of Antifungal Activity of Chitinase Proteins against Fusarium 

In the disc-diffusion susceptibility assay of the chitinases against *F. oxysporum* and *F. graminearum*, plant pathogenic fungi, at the concentration of 20 or 40 µg of XnChi, XhChi, and PtChi, most inhibition of mycelium formation showed at a concentration of 40 µg of the chitinases (Figure 6). In particular, XnChi led to the formation of a clear inhibition zone against *F. oxysporum* even at 20 µg (Appendix A). The conidial germination rates of *F. graminearum* and *F. oxysporum* were measured for 4 h or 8 h with 50, 100, 200, and 400 µg of XnChi, XhChi, and PtChi. The highest inhibition of conidial germination of both fungi was observed at the concentration of 400 µg for 8 h (Appendix A), and the growth inhibition of conidial germ tubes is shown in Figure 7. However, the antifungal activity of XnChi, XhChi, and PtChi was different depending on fungus even at the same concentration. The IC_50_ value of XnChi, XhChi, and PtChi against *F. graminearum* were 0.055 mg/mL, 0.040 mg/mL, and 0.133 mg/mL, respectively, compared with 0.072 mg/mL, 0.031 mg/mL and 0.046 mg/mL against *F. oxysporum*, respectively (Table 2).

## 3. Discussion

Chitinases identified from various bacteria have insecticidal and antifungal activities. These pesticidal chitinases have been also identified from EPN symbiotic bacteria, *Xenorhabdus* and *Photorhabdus* [32,33,40,42,43]. *Xenorhabdus* and *Photorhabdus,* having insecticidal activity, exhibit different pathogenicity depending on the species or strain [37,39]. Symbiotic bacterial chitinases may also be involved in different insecticidal activities against target pests, although they have the same functions in biological activities. Thus, chitinases of *X. nematophila*, *X. hominickii*, and *P. temperata* isolated from Korean indigenous EPNs, *S. carpocapsae* GJ1-2, *S. monticolum* GJ11-1, and *H. megidis* GJ1-2, respectively, were identified and compared to determine gene characterization using multiple sequence alignment and phylogenetic analysis. These bacterial chitinases, XnChi, XhChi, and PtChi, showed significant differences in the GH18 catalytic domain and chitinase insertion domain (CID). However, active site and chitin binding site (CBDs) were conserved (Figure 1). CBDs play an important role in interacting with insoluble chitin and promoting microbial adhesion to chitin for subsequent degradation, while CID promotes orientation and binding of longer carbohydrate substrates [24].

In the phylogenetic analysis, as shown in Figure 2, XnChi, XhChi, and PtChi were divided into distinct clades from previously studied *Pseudomonas*, *Bacillus*, and *Serratia* [44,45,46,47,48,49]. Out of three bacterial chitinases, XnChi was closer to PtChi rather than XhChi. This may be resulted from the amino acid sequence similarity (XnChi vs. PtChi = 77.47%; XnChi vs. XhChi = 52.82%). In particular, XnChi is relatively close to the *X. nematophila* ATCC19061 chitinase (Genbank accession number: WP 010846090.1), which shows 99.69% homology to Chi70 (Genbank accession number: GK44779.1) of *X. nematophila*. Chi70 was reported to have insecticidal and antifungal activities against certain pests and fungi, and to improve insecticidal activity of BtCryAc and Tc toxins [23,31,32]. Thus, symbiotic bacterial chitinases of EPNs could be promising control agents of insect pests and plant pathogenic fungi by playing an important role in pesticidal activity. However, even though *Xenorhadus* and *Photorhadus* have multiple insect-model host-specific pathogenicity, different pathogenicity factors between *Xenorhadus* and *Photorhadus* have not yet been investigated [37,39]. Thus, to compare the insecticidal activity of their chitinases from different strains within EPN species, the chitinases of *X*. *nematophila*, *X*. *hominickii*, and *P*. *temperata* were expressed and purified. The molecular weights of three recombinant chitinases were higher (Figure 3) than the predicted weight, and this might be caused by O-glycosylation, one of the post-translational modifications of the chitinase protein that occurs in some bacterial enzymes [50,51].

In the results of measuring chitinolytic activity of purified recombinant chitinases through two chitoligosaccharide analogues, endochitinase activity degrading the trimer substrate [MU-(GlcNAc)_3_ ] and chitibiosidase activity degrading the dimer substrate [MU-(GlcNAc)_2_] were observed (Figure 4). The fact that chitinases of bacterial origin exhibit both endochitinase activity and exochitinase activity, respectively, has been well-established in previous studies [52,53]. However, for Chen et al., after the study of [54], in the results of recent studies [55,56] show both endochitinase activity and exochitinase activity, similar to our chitinolytic assay results, and which are consistent with our results. Furthermore, Mahmood et al. [40] also demonstrated that chitinase from *X. nematophila* exhibits high β-N-acetylglucosaminidase activity and endochitinase activity, compared to chitiobiosidase activity. In general, previous studies have reported that exochitinase is not very efficient in chitin degradation because access to substrates is limited [57]. Therefore, the combination of exo- and endo-chitinase activity is typically several times more potent than single activity [58], implying that these chitinases may be more effective in pest control compared to chitinases from other sources. However, activity of β-1,4,N-acetylglucosaminidase, which degrades the monomer substrate [MU-GlcNAc], was not observed in XnChi, XhChi, and PtChi. This has also been observed in the measurement of the activity of metagenome-sourced chitinase Chi18H8 [55,59]. In particular, XhChi showed higher chitobiosidase activity, whereas endochitinase activity was higher in XnChi and PtChi. This might be due to different biological activities due to a difference of chitinolytic activity. Furthermore, the different amino acid sequences of the three chitinases suggest that they may have affected the structure of the substrate binding pocket/gap, which will require protein structure analysis [60].

In a previous study, a novel chitinase from *X. nematophila* was found to exhibit oral insecticidal activity against *Helicoverpa armigera* [40]. In this study, the oral insecticidal activity of XnChi, XhChi, and PtChi against *G.mellonela* was compared. We found that the insecticidal activity of three chitinases was observed in all of the developmental stages of *G. mellonella* (Appendix A), especially showing that XnChi has the highest efficacy against *G. mellonella* (Table 1). This suggests that external chitinases disrupted the controlled degradation of chitin, playing an essential role in the molting and pupal processes, which are the growth and developmental processes of insects [61,62]. The difference in virulence activity is probably as a result of their chitinase activity depending on the chitin component of the target insect as a substrate. This is consistent with the chitinolytic activity results showing that the endochitinase activity (Figure 4), which possesses the ability to cleave all parts of the chitin polymer it comes in contact with, is most prominent in XnChi. The insecticidal activity of endochitinases against plant pests has been well reported previously [30,63].

Chitin is also a major component of most fungal cell walls, so enzymes in the chitinolytic system play an important role in controlling fungi. The enzymatic lysis of the fungal cell wall via extracellular chitinase is implicated as a biological control mechanism by bacterial agents [64]. Thus, antifungal activity of three chitinases against *F. oxysporum* and *F. graminearum* was compared through disc-diffusion susceptibility assay and a conidial germination test. Inhibition of mycelial growth was observed for both fungi (Figure 6), with the size of the inhibition zone being larger for *F.oxysporum* compared to *F.graminearum*. Previous studies have reported that, even within the *Fusarium* genus, differences exist in the cell wall components and contents. *F. graminearum* contains a higher quantity of N-acetylglucosamine, phosphorus, and minerals (ash), which positively contribute to the safety and strength of the cell wall compared to *F. oxysporum* [65,66]. These results suggest that *F. oxysporum* may have been more sensitive to chitinase than *F. graminearum*. Previous studies have also shown that even within the same *Fusarium* genus, the degree of mycelial growth inhibition is different [67,68]. Inhibition of conidial germ tube elongation (Figure 7) of both fungi were observed. Out of three chitinases, XhChi showed the highest antifungal activity against *F. oxysporum* and *F. graminearum*, followed by XnChi and PtChi (Table 2). This suggests that chitinases with distinct amino acid sequences may influence specific activities based on the chitin composition of the target fungi. These results could also be inferred from Figure 4, which shows that although XhChi showed slightly lower endochitinase activity than PtChi and XhChi, chitobiosidase activity was significantly higher in XhChi than XnChi and PtChi. The result is assumed to be consistent with previous research, which reported that the combination of endochitinase and chitobiosidase showed higher antifungal activity [58]. In fact, antifungal activity of chitinases from symbiotic bacteria of EPN has been reported in several studies [32,33]. For example, the partially purified chitinase enzyme from *X. bovienii* has exhibited strong antifungal activity against *Botrytis cinerea* by inhibiting conidial germination and germ tube elongation or lysing the germ tube [54]. Additionally, the rapid increase in the chitinase activity during the first 24–48 h of bacterial culture suggests that these chitinases may play an important role in early protection against fungal invasion of dying insects [54].

The entomopathogenicity-related genes with large molecular weights (i.e., Mcf, Tc, Xpt, PVC) have been well studied in *Xenorhabdus* and *Photorhabdus* [69,70,71], but enzymatic toxins, including chitinases have been rarely studied. Furthermore, the antifungal activity of chitinases have been little studied, except for *X. nematophila* and *P. luminescence* [32,33]. In addition, studies comparing the activity of pathogenicity factors among EPN symbiotic bacteria have not yet been investigated. Therefore, this study suggest that even if they have similar biological activity, these chitinases, depending on different strains, may exhibit different pathogenicity activity against target insect and plant pathogenic fungi. Different biological activities of symbiotic bacterial chitinases indicate that these chitinases can be promising candidates for the development of pest-resistant crops [72]. Thus, further studies on the production of insect-resistant and fungus-resistant transgenic crops with EPN symbiotic bacterial chitinases and mass-production of chitinase proteins as bio-control agent are recommended.

## 4. Materials and Methods

### 4.1. Selection of Type Strain to Identify Chitinase Gene of EPN Symbiotic Bacteria

Chitinase genes of type strains were selected to identify those of symbiotic bacteria *X. nematophila*, *X. hominickii*, and *P. temperata* isolated from Korean indigenous EPNs *S. carpocapsae* GJ1-2, *S. monticolum* GJ11-1, and *H. megidis* GJ1-2, respectively.

Chitinase genes of *X. nematophila* ATCC19061 and *P. temperata* J3 were selected because not only does chitinase of *X. nematophila* ATCC19061 (GeneBank accession number: WP010846090.1) have insecticidal properties [40] but also that of *P. temperata* J3 (GeneBank accession number: WP023045972.1) has the highest similarity with *X. nematophila* ATCC19061 in *P. temperata* strains by the NCBI blastp program (https://blast.ncbi.nlm.nih.gov/Blast.cgi?PROGRAM=blastp&PAGE_TYPE=BlastSearch & LINK_LOC=blasthome, accessed on 20 November 2023).

These chitinase of two bacteria belong to the Glycoside Hydrolase Family 18 (GH18). Thus, chitinase of *X. hominickii* ANU1 belonging to GH 18 (GeneBank access number WP: 069316843.1) which is the only known strain of *X. hominickii* group in the NCBI database was used as the type strain of studied *X. hominickii* (Appendix A).

### 4.2. Chitinase Sequence Multi-Alignment and Phylogenetic Analysis

GH 18 catalytic domain, chitinase insertion domain, active site, and chitin binding site of chitinase of *X. nematophila*, *X. hominickii*, and *P. temperata* were identified through the Expasy program website (https://prosite.expasy.org/, accessed on 20 November 2023) and InterPro website (https://www.ebi.ac.uk/interpro/, accessed on on 20 November 2023) to compare amino acid sequence. The aligned chitinase amino acid sequences obtained through the MultAlign website (http://multalin.toulouse.inra.fr/multalin/, accessed on 20 November 2023) were visualized using a GeneDoc program for multiple sequence alignment analysis (Appendix A). A phylogenetic tree was constructed using the Maximum Likelihood method (bootstrap test 1000 replicate) in MEGA X bioinformatics tool. Amino acid sequences of the GH 18 family of chitinases used in the analysis were acquired from the PROTEIN category of the NCBI database (Appendix A).

### 4.3. Construction of Recombinant Chitinase-Encoding Genes

Genomic DNA of *X. nematophila*, *X. hominickii*, and *P. temperata* was extracted using DNeasy Ultra Clean Microbial Kit (Qiagen, Hilden, Germany). The primers were designed with CDS sequences of three type strains selected previously to identify a chitinase gene from genomic DNA of the above three bacteria. In addition, specific restriction enzymes were linked to the front and rear sequences of the primer for cloning (Appendix A). TOPO™ XL-2 Complete PCR Cloning Kit (Invitrogen, Carlsbad, CA, USA) was used to clone chitinase gene by performing PCR with Platinum™ SuperFi™ Green PCR Master Mix [2X] (Invitrogen, Carlsbad, CA, USA) under the following conditions: 98 °C for 30 s, 30 cycles of 98 °C for 10 s, 61 °C for 10 s, 72 °C for 1 min, and 72 °C for 10 min. The PCR products were purified using the QIAquick PCR Purification kit (Qiagen, Hilden, Germany) and cloned into pCR-XL-2-TOPO™ Vector (Invitrogen, Carlsbad, CA, USA). Then, One Shot™ OmniMAX™ 2 T1R Chemically Competent *E. coli* (invitrogen, Carlsbad, CA, USA) was transformed. Plasmids were extracted using QIAprep spin mini prep kit (Qiagen, Hilden, Germany) and sequencing was performed using Genetic analyzer 3730 xl (Applied Biosystems, Waltham, MA, USA). For the comparison of protein expression level of chitinase, a pCold II vector protein expression (TaKaRa Bio, Shiga, Japan) system was used. Topo vector-cloned plasmids were digested by *Sac*Ⅰ (TaKaRa Bio, Shiga, Japan) and purified using QIAquick PCR Purification Kit. After that, the products were digested with each of *Xba*Ⅰ, *Sal*Ⅰ, and *Pst*Ⅰ (TaKaRa Bio, Shiga, Japan) (Appendix A) restriction enzymes and gel electrophoresis was performed on 0.8% agarose gel and purified through a QIAquick Gel extraction kit (Qiagen, Hilden, Germany). Similarly, pColdⅡ vector was also double digested and purified to prevent re-ligation. A DNA ligation kit (TaKaRa Bio, Shiga, Japan) was used to subclone chitinase pcr fragments into pColdⅡ vector (TaKaRa Bio, Shiga, Japan). The final volume was 10 μL (DNA solution:Ligation Mixture = 1:1) at a ratio of 3:1 (insert: vector molar ratio). To increase the ligation efficiency after heating at 65 °C and cooling on ice for 2–3 min, overnight incubation was performed at 16 °C. The ligation product was transformed into 100 μL of DH 5α Chemically Competent *E. coli* (Enzynomics, Daejeon, Republic of Korea), and the product was screened with a selective Difo^TM^ LB Broth Miler (Bectone, Dickinson and Company, Sparks, MD, USA) medium containing 100 μg/mL ampicillin (MB cell, Seoul, Republic of Korea). Finally, the plasmids were extracted from the selected colony and transformed into a protein expression *E. coli*, BL21 (TaKaRa Bio, Shiga, Japan).

### 4.4. Expression and Purification of Recombinant Chitinase Protein

A single colony of BL21-transformant was picked up to express pCold-chitinase construct and cultured in LB medium including 100 μg/mL ampicillin for 16 to 20 h at 37 °C with a shaking incubator at 225 rpm. Then, approximately 15% of the culture was added to fresh LB medium containing 100 μg/mL ampicillin and cultured for 2 to 3 h at 37 °C in a shaking incubator until OD_600_ = 0.4 to 0.6. After being cold-shocked on ice for 30 min, cultures added with 1 mM isopropyl-β-D-thiogalactopyranoside (Thermo Scientific, Lithuania, Italy) were incubated at 15 °C in a shaking incubator to induce protein expression. The cultures were collected to confirm the level of protein expression over time at 3, 6, 9, 12, 18, and 24 h after IPTG addition using M 15R centrifuges (Hanil, Daejeon, Republic of Korea) at 13,000 rpm at 4 °C. In the control, cultures were collected at the same time without adding IPTG. The collected bacterial cells were washed twice in 1 × PBS (Bioneer, Daejeon, Republic of Korea) and mixed with 1 × FastBreak Cell Lysis Reagent (Promega, Madison, WI, USA), Lysis buffer (50 mM NaH2PO4, 300 mM NaCl, 10 mM imidazole, pH 8.0), and 0.1 mg/mL Lysozyme Solution (Thermo Scientific, Rockford, IL, USA). These were incubated in a 37 °C shaking incubator for 30 min. After that, cell mixture was subjected to cell lysis using a 21 G syringe (Koreavaccine, Seoul, Republic of Korea), and the cell lysate was centrifuged (Hanil, Daejeon, Republic of Korea) at 13,000 rpm for 30 min at 4 °C. The supernatants of cell lysates were incubated with Ni-NTA Superflow (Qiagen, Hilden, Germany) at 4 °C for 1 h at 225 rpm in a shaking incubator. The soluble fraction was dispensed onto a disposable polypropylene column (Thermo Scientific, Rockford, IL, USA) and subjected to Ni^2+^-NTA affinity chromatography through an imidazole-dependent gradient method (Qiagen manual) as follows; the loaded suspension was washed twice with wash buffer (50 mM NaH_2_PO_4_, 300 mM NaCl, 10 mM or 20 mM imidazole, pH 8.0), and proteins were eluted with elution buffer (50 mM NaH_2_PO_4_, 300 mM NaCl; 100 mM, 150 mM or 200 mM or 300 mM imidazole; pH 8.0). Then, 2 × sample buffer (Bio-Rad, Hercules, CA, USA) supplemented with 2- Mercaptoethanol (Bio-Rad, Hanghai, China) and the purified protein were mixed at a ratio of 1:1. The mixture was boiled using a Microprocessor block heater (Barnstead/Lab-LINE, Melrose park, IL, USA) for 5 min, followed by cooling on ice for 2 min and centrifuging at 13,000 rpm at 4 °C. Finally, 20 μL of the supernatant was loaded onto a 4–15% SDS–PAGE (Bio-Red, Hercules, CA, USA). After electrophoresis, staining and destaining were performed through Coomassie Brilliant Blue R—250 Solution (Ezynomics, Daejeon, Republic of Korea) and Coomassie Brilliant Blue R—250 destaining Solution (Ezynomics, Daejeon, Republic of Korea), respectively. Some 30 kDa Ultra-15 centrifuge Filter Devices (Amicon, Darmstadt, Germany) were also used for the concentration and desalting of protein. After pre-rinsing filter devices with Ambion DEPC-treated Water (Invitogen, Waltham, MA, USA), purified protein was added and centrifuged 3500× *g* for 20 min at 4 °C in Combi R 515 Swinging bucket rotor (Hanil, Daejeon, Republic of Korea). Subsequently, the filtrate was discarded, and we added DEPC-treated water. After that, centrifugation was performed under the same conditions as above. After washing twice, the protein concentrated in the filter was recovered and quantified manually using the Pierce BCA Protein Assay kit (Thermo Scientific, Rockford, IL, USA).

### 4.5. Chitinolytic Activity Assay with Recombinant Chitinase Protein

The activity of purified chitinase was measured using a chitinase assay kit, fluorimetric (Sigma-Aldrich, St. Louis, MO, USA). The enzyme activity of chitinase was measured by detecting the fluorescence of 4-methylumbelliferone (MU), a fluorescent dye generated during the hydrolysis of chitooligosaccharide analogs, which are substrates of chitinase: 4-Methylumbelliferyl β-D-N,N′,N″-triacetylchitotriose [4-MU-(GlcNAc)_3_], 4-Methylumbelliferyl N,N′-diacetyl-β-D-chitobioside [4-MU-(GlcNAc)_2_], and 4-methylumbelliferyl N-acetyl-β-D-glucosaminide [4-MU-GlcNAc]. The emission of 4-methylumbelliferone (MU) was measured by fluorescence detection with a PerkinElmer 2030 multilabel plate reader (PerkinElmer, Waltham, MA, USA) under the conditions of pH 5.0 and 360 nm excitation wavelength and 450 nm emission wavelength. 4-Methylumbelliferyl N,N′-diacetyl-β-D-chitobioside [4-MU-(GlcNAc)_2_] and 4-methylumbelliferyl N-acetyl-β-D-glucosaminide [4-MU-GlcNAc] were used to measure exochitinase activity, while 4-Methylumbelliferyl β-D-N,N′,N″-triacetylchitotriose [4-MU-(GlcNAc)_3_] was used to measure endochitinase activity. Simultaneously, 0, 25, 100, 500, 800, and 1000 ng/µl of purified chitinase protein were added to MicroWell™ 96-well black plates (Sigma Aldrich, Roskilde, Denmark) to assess concentration-dependent substrate degradation with the above three substrates. After 30 min incubation of the above mixture at 37 °C, Stop Solution (Sodium Carbonate) was added to each well to measure substrate resolution.

### 4.6. Insecticidal Activity Bioassay of Recombinant Chitinase Protein against Galleria mellonella

Insecticidal activity of purified chitinase proteins of three symbiotic bacteria was evaluated against *G. mellonella* using oral feeding assay on 1 g artificial diet (Rice bran 33.1%, Wheat bran 33.1%, Yeast Extract 0.2%, Calcium propionate 0.6%, Honey 13.2%, Glycerin 13.2%, Vitamin 0.05% and 6.65%) mixed with various concentrations (0.8, 0.4, and 0.1 mg/mL) of purified chitinase protein dissolved in 0.5 mL of deionized purified water (DDW) in 60 mm Petri dishes for 2nd to 3rd instars and 100 mm Petri dishes for 4th instar. Larvae in the control were fed on only an artificial diet treated with the same volume of DDW. The artificial diet treated with DDW and chitinase proteins was dried on a clean bench for one day before use.

Ten 24 h-fasted 2nd instar larvae were placed in each Petri dish and reared in environmentally controlled conditions at 26 ± 2 °C and 69% relative humidity. Larval mortality was checked every day, and food was replaced as needed. The experiment was performed in triplicate using ten larvae per replicate. Larval mortality depending on the concentration of each bacterium was compared by the by as a post hoc Turkey’s test, and LT_50_ values were calculated using probit analysis through IBM SPSS Statistics 27 software program.

### 4.7. Antifungal Activity Assay of Recombinant Chitinase Protein against Plant Pathogenic Fungi

The antifungal activity of purified chitinase proteins of three symbiotic bacteria was evaluated against plant pathogenic fungi *F*. *graminearum* EZ 3639 and *F*. *oxysporum* by conidial germination test and disc-diffusion susceptibility assay. The fungi used in the test were provided from the Fungal Plant Pathology Laboratory, Dong-A University, Busan, Republic of Korea. Mycelial blocks of fungi cultivated on potato dextrose agar (PDA) for 3 days were inoculated into carboxyl methyl cellulose (CMC) medium [73] to induce conidia and were cultured at 200 rpm at 25 °C for 5 days [74]. Cultivated conidia were collected using centrifugation at 13,000 rpm at 4 °C for 30 min, and then washed twice using DDW. Chitinase proteins diluted serially with concentrations of 0.05, 0.1, 0.2, and 0.4 mg/mL (final volume of 1 mL) were inoculated into 25 mL of potato dextrose broth (PDB) adjusted to a final concentration of 2 × 10^5^ conidia/mL. DDW of the same volume was used as the control. The ratio of germinated conidia was counted using microscopy (Nikon ECLIPSE Ci, India) at 4 and 8 h after inoculation. The presence or absence of conidial germination was judged to indicate germination when fungi reached more than 1/2 of the spore length. The IC_50_ values of conidia inhibition rate were calculated using probit analysis and germination rate (germinated conidia/total conidia) for the chitinase of each bacterium was compared with a post hoc Turkey’s test using the IBM SPSS Statistics 27 software program [74]. In the disc-diffusion susceptibility assay, fungal conidia were collected in the same way as above for measuring the conidial germination rate. The collected conidia were inoculated into a sterilized PDA medium at a final concentration of 2 × 10^4^ conidia/mL, and then were cultured at 25 °C for one day to solidify the medium. Finally, 40 µg (final volume 40 µL) of purified chitinase protein was dropped into a spore-exposed medium, dried for 5 min, cultured at 25 °C, and observed on the next day. An equal volume of DDW was used as a negative control.

## Figures and Tables

**Figure 1 toxins-16-00026-f001:**
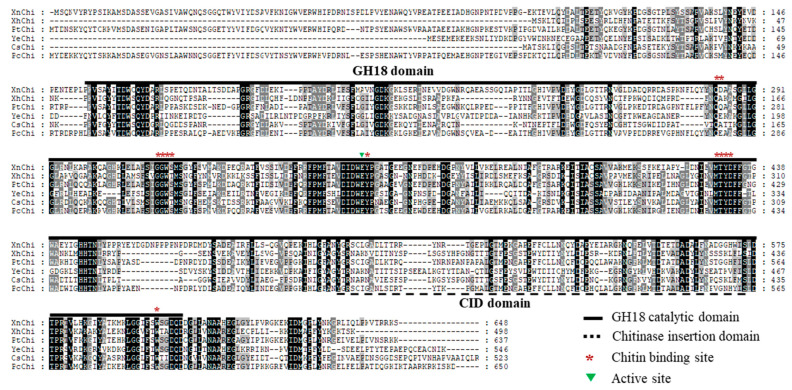
Multiple sequence alignment for the amino acid sequences of chitinase. The multiple sequence alignment comparison analysis of chitinase amino acid sequences including XnChi, XhChi, PtChi, and three other GH18 chitinases, was performed through the GeneDoc program. Identical and similar residues are shaded black and shaded gray, respectively. The GH18 domain is represented by a bold black line, CID domain is represented by a black dotted line, chitin binding site is represented by a red asterisk, and active site is represented by a green triangle. YeChi, CsChi, and PcChi represent their respective chitinase for *Yersinia entomophaga* MH96 (GeneBank accession number: ANI28952.1), *Cronobacter sakazakii* wls2261 (GeneBank accession number: WP241700376.1), and *Pseudomonas chloriraphis* B25 (GeneBank accession number: WP124321822.1).

**Figure 2 toxins-16-00026-f002:**
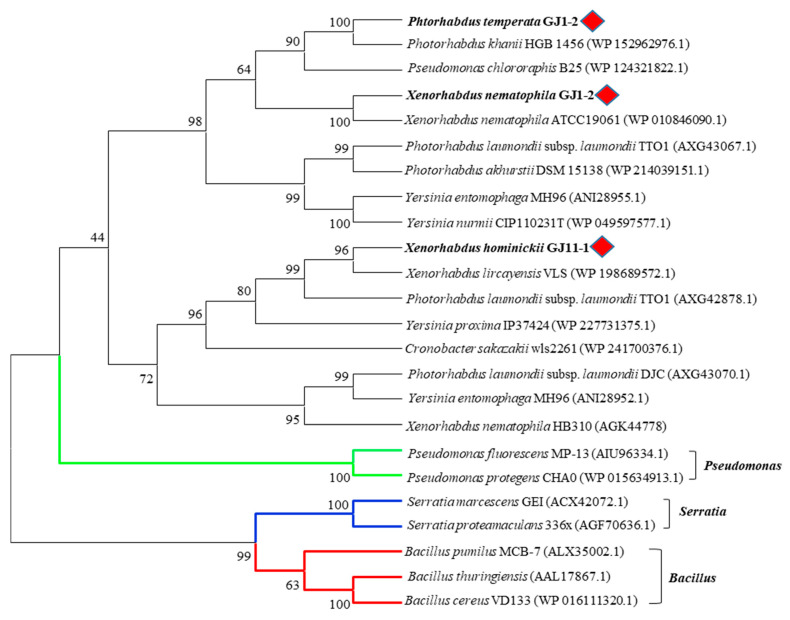
Phylogenetic analysis of various bacterial chitinases. The phylogenetic tree was constructed using a Maximum Likelihood method based on the amino acid sequence alignment in MEGA X program. The aligned GH18 family chitinase amino acid sequence was obtained from the NCBI database. The red rhombus represents XnChi, XhChi, and PtChi used in this study and green, blue, and red lines indicated *Pseudomonas* spp., *Serratia* spp. and *Bacillus* spp., respectively. The percentage of replicate trees in which the associated taxa clustered together in the bootstrap test (1000 replicate) are shown next to the branches. All positions containing gaps and missing data were eliminated (complete deletion option).

**Figure 3 toxins-16-00026-f003:**
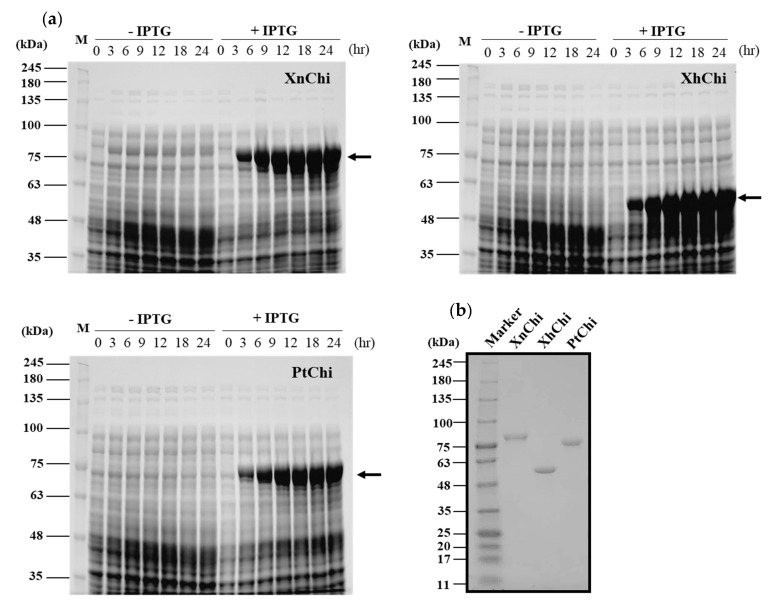
Expression and purification of the recombinant XnChi, XhChi, and PtChi. (**a**) A total of 10% SDS–PAGE analysis of chitinase protein induced by 1 mM IPTG through sampling for 3 h, 6 h, 9 h, 12 h, 18 h, and 24 h, respectively. From the left, XnChi, XhChi, and PtChi are shown. (**b**) Chitinase protein purified using Ni–NTA resin was concentrated and desalted using an Amicon tube (30 kDa-cut off). Concentrated chitinase protein (2 µg) was analyzed using 4–15% SDS–PAGE. Lane M: Protein marker. The protein bands corresponding to chitinase protein are indicated with an black arrow.

**Figure 4 toxins-16-00026-f004:**
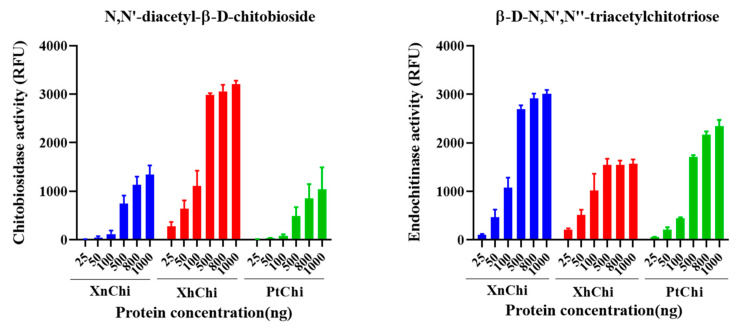
Chitinolytic activity of the recombinant XnChi, XhChi, and PtChi. Enzyme activity of chitinase using two substrates known as chitoligosaccharide analogs: 4-methylumbeliferyl N,N′-diacetyl-β-D-chitobioside [4-MU-(GlcNAc)_2_] and 4-Methylumbelliferyl β-D-N,N′,N″-triacetylchitotriose [4-MU-(GlcNAc)_3_]. The kinetic curve for the degree of hydrolysis of the substrate (0.5 mg/mL at the final volume of 100 μL) was measured according to the standard curve prepared using fluorescence readings of five standard solutions. The different concentrations of the samples used were 25, 50, 100, 500, 800, and 1000 ng/μL.

**Figure 5 toxins-16-00026-f005:**
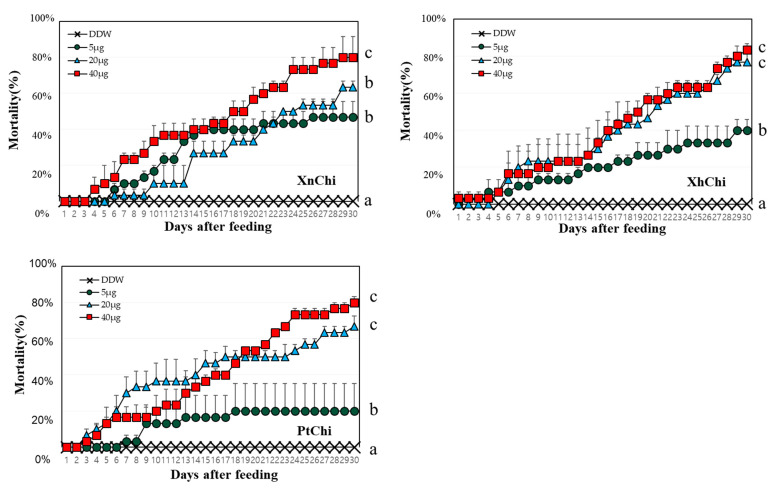
Mortality according to concentration of chitinases against *G. mellonella*. 10 of second larval instar reared on different concentration of protein (5, 20 and 40 µg/larva) for 30 days. Mortality was calculated as the number of dead larvae/total number of larvae. Deionized purified water (DDW) was used as control. Turkey’s test was used as a post hoc method to identify the concentration of high insecticidal activity that showed significant differences. Values within a figure with different letters are significantly different according to the Tukey’s test (*p* < 0.05) in the IBM SPSS Statistics 27 software program.

**Figure 6 toxins-16-00026-f006:**
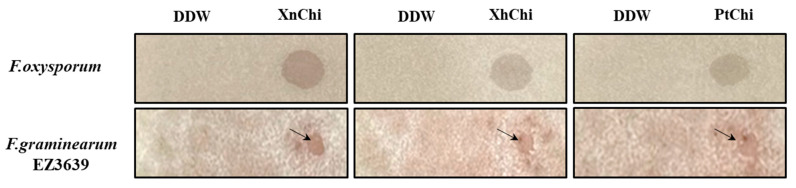
Antifungal activity of XnChi, XhChi, and PtChi (40 µg) against *F. oxysporum* and *F. graminearum*. Inhibition of mycelium growth by chitinase was shown as the inhibition zone through disc-diffusion susceptibility assay. No growth inhibition was observed in the control. The arrow indicates the zone of inhibition for mycelium growth of the fungi.

**Figure 7 toxins-16-00026-f007:**
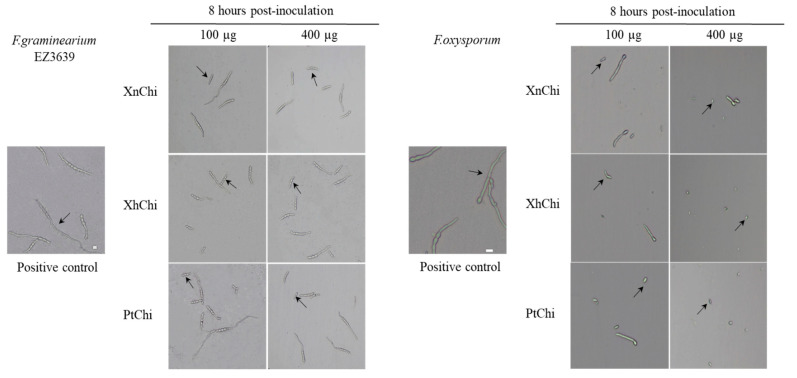
Effect of XnChi, XhChi, and PtChi on conidial germination of *F. graminearum* and *F.oxysporum*. Chitinases were added to Potato dextrose broth (PDB) at 100 µg and 400 µg concentrations, and conidial germination was measured at 8 h after inoculation with 2 × 10^5^ conidia/mL. Fungal inoculum was used as positive control. The germinated conidia were observed through light microscopy. The germinated conidia were judged to have grown more than 1/2 of the conidial length. The scale bar was measured through the image J program. Scale bar = 10 μm. The arrow indicates the difference of conidia between the treatments and control.

**Table 1 toxins-16-00026-t001:** LT_50_ and LT_70_ values of orally administered XnChi, XhChi, and PtChi against larvae of *G. mellonella*.

Protein	Time to Death and/or Death Response (40 μg/Diet-Fed Larva)		
	LT_50_ (days)	LT_70_ (days)	R^2^ Value
XnChi	16.155 (14.066–18.661)	25.289 (21.543–31.838)	0.964
XhChi	18.337 (15.879–21.666)	29.433 (24.460–38.900)	0.880
PtChi	17.797 (15.384–21.307)	27.841 (22.944–37.917)	0.932

Values were determined via probit analysis using an IBM SPSS program. Numbers in parentheses represent 95% confidence limits. The LT_50_/LT_70_ values were considered significantly different if the 95% confidence interval (CI) values did not overlap with the CI values of other treatment. The R^2^ (regression coefficient) is indicative of the closeness of data to the fitted regression line.

**Table 2 toxins-16-00026-t002:** Comparison of antifungal activity of XnChi, XhChi, and PtChi against *F. graminearum* and *F. oxysporum*.

Protein	Toxicity in mg/mL to *F. graminearum* EZ3639		
	8 h		
	IC_50_	IC_70_	R^2^ Value
XnChi	0.055 (0.015–0.089)	0.325 (0.201–1.431)	0.987
XhChi	0.040 (0.010–0.068)	0.200 (0.135–0.392)	0.956
PtChi	0.133 (0.080–0.210)	0.742 (0.382–5.226)	0.949
**Protein**	**Toxicity in mg/mL to *F. oxysporum***		
	**8 h**		
	**IC_50_**	**IC_70_**	**R^2^ Value**
XnChi	0.072 (0.042–0.098)	0.239 (0.175–0.397)	0.938
XhChi	0.031 (0.004–0.060)	0.191 (0.122–0.423)	0.971
PtChi	0.046 (0.024–0.066)	0.609 (0.378–1.562)	0.931

The IC_50_ value was calculated through the conidial germination inhibition rate. Conidial germination inhibition rate was observed at 8 h after inoculation. The IC_50_ and IC_70_ values of chitinase, treated at a concentration of 0.4 mg/mL against *F. graminearum* (upper) and *F. oxysporum* (lower) with 2 × 10^5^ conidia/mL, respectively, were determined.

## Data Availability

Data are contained within the article or Appendix A.

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
