# Peer review of "Functional Comparison of Three Chitinases from Symbiotic Bacteria of Entomopathogenic Nematodes"

_toxins, 2024, doi:10.3390/toxins16010026_

Round 1

Reviewer 1 Report

Comments and Suggestions for Authors

Dear Authors,

Interesting topic of work and promising results with application potential. I'm also interested in the results concerning the antifungal activity of the purified enzymes and what caused the differences between the results obtained for these two fungi of the genus Fusarium? Overall the paper is quite well written, although the text needs careful proofreading by a competent person in the field of biology. 

Comments on the Quality of English Language

Some sentences are not understandable or misleading, e.g. :'These two bacteria belong to the Glycoside Hydrolase Family 18 (GH18)' verse 350

Author Response

We are pleased and grateful to receive your feedback on this manuscript. We appreciate your recognition of the interesting research topic and promising results. We will carefully consider the reviewer's comments and try to improve the manuscript until we submit a revised version in the future. Please find the detailed responses below and the corresponding revisions/corrections highlighted/in track changes in the re-submitted files.

Reviewer 2 Report

Comments and Suggestions for Authors

Dear Authors,

The comparative analysis of enzymes, their functional and ecological characterization is a challenging task. I appreciate the effort you put into these investigations. Moreover, such data might enable chemical-reduced pest management in the future. Thus, your study will possibly contribute to more environment-friendly control of crop pests. Content-wise the introduction is sufficient to get the rational of the study. However, language polishing as well as more precise explanations of statements are essentially needed to better understand your point. I highlighted some parts in blue throughout the manuscript where English polishing is required. Here are some examples where statements need further explanation.

Lines 56 and 71: Please, make sure you are introducing abbreviations correctly (Bt and GH).

Line 85: “show high similarity between clades”; what kind of similarity do you mean?

Line 107: “sequence analysis”; what kind of sequence analysis?

Line 113: “GH18 catalytic domain”; DxxDxDxE is usually the conserved motif. The first “D” is missing in your bacterial counterparts. Any comment on that?

Line 149: “chitobiosidase”; Please, indicate this as exo-activity as early as possible. People outside the field will be los otherwise.

Line 158: “LT50”; needs to be explained.

I uploaded a commented version of your manuscript together with my report. Therein you will find further examples.

In addition, I want to highlight three major issues that need to be addressed.

First, I cannot see an inhibition zone in Fig.6 (F. graminearum). I am not used to these assays, but compared to the obvious halos in case of the other Fusarium species, there is hardly anything to see in F. graminearum. Please, comment on that.

Secondly, I had difficulties to understand the distinction between exo- and endo-activity based on the substrates you used. The same holds true for some studies that I found while screening the literature. Is a trimer really a suitable substrate to reveal endo-activity? An exo-chitinase may remove the monomers stepwise and the 4MU will be released at the end as well. Although I am working on completely different GH families, I believe a polymeric substrate like colloidal chitin would be better. Please, correct me if I am wrong. I am happy to read your explanation.

Third, I found it puzzling to see that one of your enzymes, XnChi, has been already characterized in a previous study (reference 37). Two questions: Why did you include this enzyme again? And even more confusing: why did you get different results compared to ref37? β-N-acetylglucosaminidase activity was found in ref37 but not in your study and vice versa in case of chitobiosidase activity.

I am happy to read a revised version of your manuscript,

Kind regards

Comments on the Quality of English Language

Please, find my detailed comments on English polishing in my suggestions as well as directly in the commented version of the manuscript (uploaded as well).

Author Response

Thank you very much for taking the time to review this manuscript. We greatly appreciate the time and effort spent by the reviewer, for the evaluation of the above-mentioned manuscript. Please find the detailed responses below and the corresponding revisions/corrections highlighted/in track changes in the re-submitted files.

Reviewer 3 Report

Comments and Suggestions for Authors

The paper concerns the characterization of chitinases as insecticidal and antifungal molecules. The paper describes very well the topic, the methods and the results.

However, I have some questions about teh phylogenetic tree: how the authors explain the Pseudomonas sequence in the Phtorhabdus clade?

Furthermore, how the authors explain the different antifungal activity against the different fungi tested?

Furthermore. I suggest to include fig s2 in teh main test, that is more informative than fig. 5. That it is nor clear at all to me.

Also fig. 6 is nor clear to me: where is the Inhibition of mycelium growth?

After these minor modifications, the paper is acceptable for publlication.

Author Response

We would like to thank the reviewer for their positive feedback. We are pleased to hear that you were satisfied with the paper's understanding of the research topic, experimental methods, and results.

Please find the detailed responses below and the corresponding revisions/corrections highlighted/in track changes in the re-submitted files.

Round 2

Reviewer 2 Report

Comments and Suggestions for Authors

Dear Authors,

congratulations for this extensive and precise response to my comments and suggestions. There is only one thing you did not take into account, which is linked to the previous study of Mahmood. In line 285ff you still mention these authors found chitobiosidase activity. Based on the original article this is not the case; at least to me. Please, clarify this. Additionally, please mention the discrepancy between your and their characterization of the enzyme. Thank you for providing your raw data and being so open! I highly appreciate that!

Kind regards

Comments on the Quality of English Language

Minor English polishing needed.

Author Response

Thank you for your positive review of our responses to your comments and suggestions. We truly appreciate your recognition and appreciate the time and effort you put into reviewing our work.
